# Visual Context Window Extension: A New Perspective for Long Video Understanding

## Abstract

Large Multimodal Models (LMMs) have demonstrated impressive performance in short video understanding tasks but face great challenges when applied to long video understanding. In contrast, Large Language Models (LLMs) exhibit outstanding capabilities in modeling long texts. Existing work attempts to address this issue by introducing long video-text pairs during training. However, these approaches require substantial computational and data resources. In this paper, we tackle the challenge of long video understanding from the perspective of context windows, aiming to apply LMMs to long video tasks without retraining on long video datasets. We first conduct an in-depth analysis of why pretrained LMMs struggle to understand lengthy video content, identifying that *discrepancies between visual and language modalities lead to different context windows for visual and language tokens*, making it difficult to directly extend the visual tokens to match the language context window. Based on this, we propose to adapt LMMs for long video understanding tasks by extending the visual context window, eliminating the need for retraining on large-scale long video datasets. To further mitigate the significant memory consumption caused by long sequences, we introduce a progressive pooling inference strategy that selectively adjusts the spatial resolution of frame embeddings, reducing the number of visual tokens while retaining important spatial information. Across multiple long video understanding benchmarks, our method consistently improves the performance as the number of video frames increases. On the MLVU benchmark, our method outperforms GPT-4o, even though our model size is only 7B. Additionally, in the 256-frame setting, our method reduces memory usage by approximately 45% compared to the baseline, without introducing any performance loss.

## 1 Introduction

Large Multimodal Models (LMMs), built on pre-trained Large Language Models (LLMs) and trained on massive image-text pairs, have shown remarkable capabilities in image understanding (Li et al., 2023b; Gao et al., 2023; Dai et al., 2023; Zhu et al., 2023; Ye et al., 2023; Li et al., 2023a; Liu et al., 2023a). Recently, by segmenting high-resolution images into multiple sub-images for input, LMMs have not only improved in fine-grained image understanding but also demonstrated zero-shot video understanding capabilities (Liu et al., 2024b; Yao et al., 2024; Li et al., 2024a). Despite these advancements, current LMMs are still limited to short video understanding tasks and face difficulties when applied to long videos due to the excessive sequence lengths involved.

Several approaches (Li et al., 2023d; Jin et al., 2023; Song et al., 2024) have explored using visual resamplers to reduce the number of visual tokens, allowing the models to process more video frames. However, this token reduction inevitably leads to a loss of critical information, negatively affecting performance. Recent efforts (Xue et al., 2024; Liu et al., 2024c) have tackled this issue by incorporating long video-text pair datasets during pre-training. However, this approach faces significant challenges due to the high computational cost associated with the quadratic complexity of the attention mechanism (Vaswani et al., 2017) and the scarcity of high-quality long video-text data.

To alleviate the high computational costs and data collection challenges associated with long video understanding, we approach the problem from the perspective of the context window. First, we observe that in recent open-source LMMs, language decoders generally support longer language

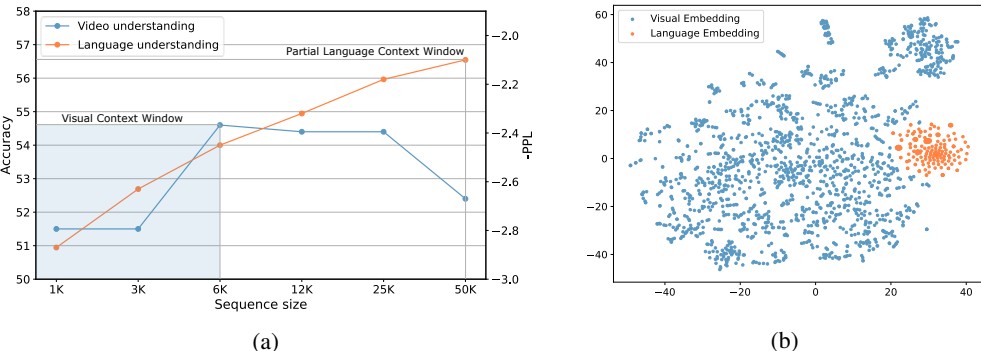

(a)                                                      (b)

Figure 1: (a) The blue curve of illustrates the accuracy comparison of different video sequence lengths on LongVideoBench (180s-600s) (Wu et al., 2024). The yellow curve shows the sliding window perplexity ($S = 256$) of ten 128k Proof-pile documents (Azerbayev et al., 2023), and for the sake of comparison, we take the negative of the perplexity. Visualization of visual embeddings (output of the modality projection layer) and language embeddings in the language decoder using t-SNE (Van der Maaten & Hinton, 2008). The visual embeddings and language embeddings form two distinct clusters.

modeling (Yao et al., 2024; Li et al., 2024a). For instance, the latest LMM, LLaVA-OneVision (Li et al., 2024a), employs Qwen2 (Yang et al., 2024) as its language decoder. As illustrated in Figure 1a, the performance of LLaVA-OneVision in language understanding tasks improves consistently as the input sequence length increases (yellow curve). However, for visual understanding tasks, the performance initially improves but then declines as sequence length grows (blue curve). Further visualization of the latent space inside the language decoder shows that visual and language embeddings form distinct clusters (Figure 1b), indicating significant modal differences in the latent space. This explains the performance of LMMs on visual understanding tasks shown in Figure 1a. We believe that due to the differences between the visual and language modalities, LMMs pre-trained on short visual sequences cannot directly extrapolate visual tokens to the effective context window size of the language decoder. Therefore, we redefine the context window in LMMs as two distinct windows: **the visual context window**, representing the maximum length of visual tokens during pre-training, and **the language context window**, referring to the maximum length of language tokens during pre-training.

Building on this observation, we propose to extend the commonly used language context window extension method, YaRN (Peng et al., 2024), to LMMs for long video understanding. Specifically, we redefine the scaling factor of the base frequency in positional embeddings as the ratio of the visual context window to the target context window. By modulating the rotational frequency of the positional embeddings, we expand the effective range of the visual context window, enabling LMMs to handle longer video sequences. It's important to note that extending the visual context window does not directly narrow the modality gap between visual and language embeddings.

Additionally, to alleviate the rapid memory consumption caused by long sequences, we propose a progressive pooling strategy to handle video frame embeddings. Specifically, considering the redundancy between consecutive frames in the same event, such as a static background, we uniformly sample the video frames into multiple groups. We assume that each group represents an event, and we control the group size through hyperparameters. In each group, the first frame's embedding retains a higher spatial resolution, while the subsequent frames are pooled with a larger stride to lower resolutions. We believe the first frame preserves rich spatial, fine-grained information compared to the other frames within the group, while the remaining frames reduce intra-group redundancy. This approach minimizes the loss of spatial information while reducing the number of visual tokens.

Across multiple long video understanding benchmarks, our method consistently improves performance as the number of video frames increases. Notably, on the MLVU benchmark (Zhou et al., 2024), our method outperforms GPT-4o. Most importantly, our approach does not require retraining, allowing it to benefit from continuous advancements in open-source LMMs.

In summary, our paper makes the following key contributions:

- We exploit the modality difference between visual and language tokens in the language decoder to redefine the effective context window in LMMs: the visual context window and the language context window.

- We propose a method to extend positional embeddings within the visual context window, enabling LMMs to handle long video tasks without the need for training on long video-text paired data.

- We introduce a progressive pooling strategy for visual frame embeddings, mitigating reducing memory consumption in long video sequences.

## 2 BACKGROUND AND RELATED WORK

### 2.1 ROTARY POSITION EMBEDDINGS

Rotary Position Embeddings (RoPE) (Su et al., 2024) introduce a rotation matrix to incorporate relative positional information into the self-attention mechanism, enhancing the model's ability to capture positional relationships between words.

Given a sequence $\mathbf{S} = \{\mathbf{w}_i\}_{i=1}^{N}$ with corresponding embeddings $\mathbf{E} = \{\mathbf{x}_i\}_{i=1}^{N}$, the query and key vectors are computed as: $\mathbf{q}_m = f_q(\mathbf{x}_m, m)$, $\mathbf{k}_n = f_k(\mathbf{x}_n, n)$, where $m$ and $n$ are positions in the sequence. The unnormalized attention scores are then calculated by dot-producting two vectors: $\mathbf{q}_m^T \mathbf{k}_n$. To incorporate relative positional information, the query and key vectors are represented in complex form:

$$f_q(\mathbf{x}_m, m) = e^{im\Theta}(\mathbf{W}_q \mathbf{x}_m), \quad f_k(\mathbf{x}_n, n) = e^{in\Theta}(\mathbf{W}_k \mathbf{x}_n), \tag{1}$$

where $\Theta = \text{diag}\left(\theta_j = b^{-2j/d}, j \in [1, 2, \ldots, d/2]\right)$ is the diagonal matrix and $b = 10000$.

In real coordinates, RoPE can be expressed using the following function:

$$f_q(\mathbf{x}_m, m) = \mathcal{R}_m(\mathbf{W}_q \mathbf{x}_m) =$$
$$\begin{pmatrix} \cos m\theta_1 & -\sin m\theta_1 & \cdots & 0 & 0 \\ \sin m\theta_1 & \cos m\theta_1 & \cdots & 0 & 0 \\ 0 & 0 & \cdots & 0 & 0 \\ 0 & 0 & \cdots & 0 & 0 \\ 0 & 0 & \cdots & \cos m\theta_{d/2} & -\sin m\theta_{d/2} \\ 0 & 0 & \cdots & \sin m\theta_{d/2} & \cos m\theta_{d/2} \end{pmatrix} \mathbf{W}_q \mathbf{x}_m. \tag{2}$$

Therefore, when the word embedding $\mathbf{x}_m$ at position $m$ is multiplied by matrix $\mathcal{R}_m$, and the word embedding $\mathbf{x}_n$ at position $n$ is also multiplied by matrix $\mathcal{R}_n$, resulting in the transformed query and key vectors, the attention weights will inherently include the relative positional information. We provide a more detailed derivation of RoPE in Appendix A.2.

### 2.2 RELATED WORK

**Large Multimodal Models** LMMs typically consist of a visual encoder, a pre-trained LLM, and a modality projection module that converts visual content into token sequences for the LLM. Leveraging large amounts of high-quality image-text paired data, LMMs have shown strong capabilities in image understanding (Li et al., 2023b; Gao et al., 2023; Dai et al., 2023; Zhu et al., 2023; Ye et al., 2023; Li et al., 2023a; Liu et al., 2023a; 2024b; Yao et al., 2024; Li et al., 2024a). By sampling videos into multiple frames, LMMs can extend to video understanding tasks (Xu et al., 2024; Chen et al., 2023a; Maaz et al., 2024; Liu et al., 2023b; Li et al., 2023c; 2024b). Examples include Video-ChatGPT (Maaz et al., 2024), VideoChat2 (Li et al., 2024b), and PLLaVA (Xu et al., 2024), which enhance LMMs' video understanding through high-quality data and fine-tuning methods. However, these methods face challenges with long videos due to the large number of visual tokens generated per frame.

To address this, visual token compression methods have been proposed (Li et al., 2023d; Jin et al., 2023; Song et al., 2024). For instance, LLaMA-VID (Li et al., 2023d) uses only two tokens per frame, and MovieChat (Song et al., 2024) introduces a memory mechanism to compress long video tokens into a fixed size. These methods, however, often result in information loss.

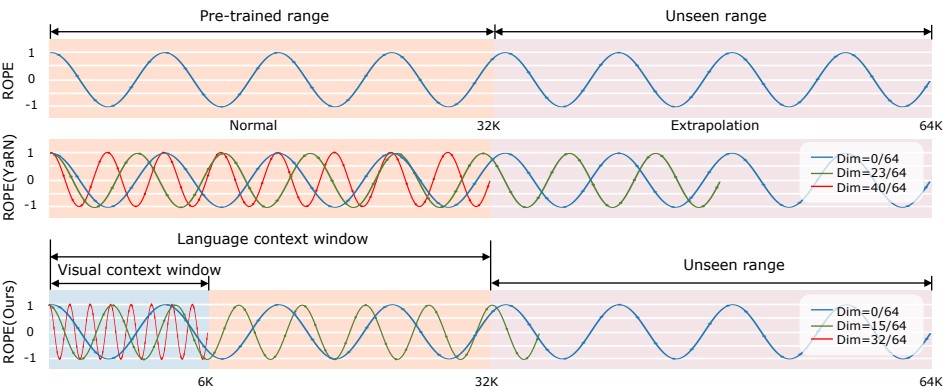

Figure 2: Examples of RoPE embeddings under different context extension methods. Upper: RoPE directly extrapolated beyond the pre-training range. Middle: YaRN interpolating and extrapolating different RoPE dimensions beyond the pre-training range. Down: Our method further distinguishes between visual and language context windows in YaRN, allowing for different interpolation and extrapolation of RoPE dimensions.

In recent work, LongVILA (Xue et al., 2024) attempted to introduce long video-text pairs into the training of LMMs to expand the context window size. LongVA (Zhang et al., 2024) expands the context window by continuously training LLMs on long texts, transferring its long text understanding capabilities to long video understanding. However, they inevitably introduce high computational costs and data collection challenges.

**Context Window Extension for LLMs** The fixed context length during pre-training limits the inference performance of language models in scenarios involving long sequence inputs. To address this issue, researchers have proposed a series of RoPE-based language positional embedding extension methods, such as Position Interpolation (PI) (Chen et al., 2023b; kaiokendev, 2023), NTK Interpolation (bloc97, 2023), and YaRN (Peng et al., 2024). Specifically, PI scales the positions of long texts that exceed the context window down to the original window size. However, it compresses distances between nearby tokens, which can degrade performance. NTK interpolation extends the context window by adjusting the rotational speed of RoPE through reducing the base frequency. Building upon NTK interpolation, YaRN further distinguishes between high-frequency and low-frequency information to accommodate different RoPE embeddings.

## 3 METHOD

In this section, we first introduce the corresponding modifications of the language position embedding extension method to the visual context window. We then further discussed another factor that limit long video understanding: memory constraints.

### 3.1 VISUAL CONTEXT WINDOW EXTENSION

In Section 2.1, we describe the commonly used position embedding method in LLMs and LMMs, RoPE (Rotary Position Embedding). LLMs typically have a fixed context window size, and when the input sequence exceeds this limit, the model struggles to accurately understand positional information, leading to a decline in performance. As shown in Figure 1a, LMMs encounter similar issues when processing long video sequences.

To address this, we adapt the language position embedding extension method, YaRN (Peng et al., 2024), for the visual context window to better support long video understanding. Figure 2 illustrates an example of our method. In our approach, we define the training context length for visual data as $L_{\text{train}}^v$ (i.e., visual context window), and the extended context length as $L_{\text{test}}^v$. Consequently, we define the scaling factor $s$ as follows:

$$s = \frac{L_{\text{test}}^v}{L_{\text{train}}^v}. \tag{3}$$

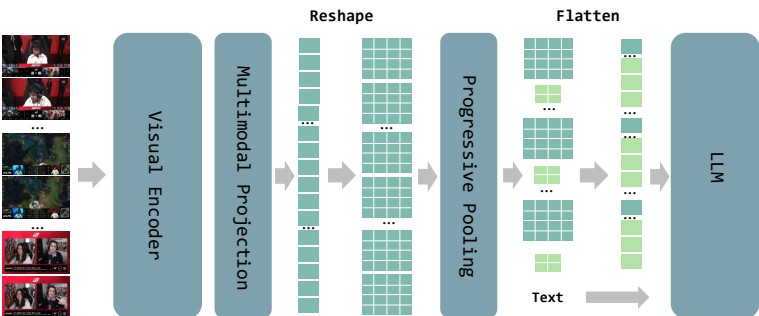

Figure 3: Pipeline of progressive pooling strategy.

Then, we selectively interpolate the hidden dimensions based on the wavelength $\lambda_i$ of the RoPE embeddings:

$$\lambda_i = \frac{2\pi}{\theta_i} = 2\pi b^{\frac{2i}{d}}. \tag{4}$$

Following this, we define $r_i = \frac{L_{\text{train}}^v}{\lambda_i}$ to determine which dimensions require interpolation. Finally, following YaRN, combining the scaling factor $s$ with the wavelength $\lambda_i$, the base frequency is modified as follows:

$$\theta_i^{\text{new}} = \left[ \gamma_i + (1 - \gamma_i) \frac{1}{s} \right] \theta_i, \quad \gamma_i = \begin{cases} 1, & r_i > \beta \\ 0, & r_i < \alpha \\ \frac{r_i - \alpha}{\beta - \alpha}, & \text{otherwise,} \end{cases} \tag{5}$$

where, $\alpha$ and $\beta$ are hyperparameters. When $r_i < \alpha$, we apply linear interpolation proportionally based on $s$. When $r_i > \beta$, no interpolation is applied. For cases between $\alpha$ and $\beta$, we apply a linear interpolation transition. We provide detailed derivations of context window extension method in Appendix A.3. It is important to note that our modifications to YaRN are minimal, ensuring simplicity and compatibility with various acceleration techniques, such as flash-attention (Dao et al., 2022).

## 3.2 PROGRESSIVE POOLING

In this section, we discuss another factor that limits the performance of long video understanding: memory constraints. Taking LLaVA-OneVision as an example, given a video $V$ uniformly sampled into $N$ video frames, the visual encoder and multimodal projection module process these frames to obtain the video sequence embeddings $F_v \in \mathbb{R}^{N \times 729 \times d}$. To reduce the number of visual tokens, LLaVA-OneVision performs bilinear pooling with a stride of 2 on each video frame embedding, which then serves as the input to the LLM decoder. However, even after bilinear pooling, a video sequence of 256 frames generates 50,176 tokens.

Long sequences contribute to high memory consumption. Inference in LMMs can be divided into two stages: prefill and decoding. During the prefill stage, all visual tokens are projected into a high-dimensional space and stored as KVCache for efficient decoding later. This incurs substantial memory costs. Even with bilinear pooling, processing 256 frames generates 50,176 tokens, requiring approximately 73 GB of GPU memory. This greatly limits the deployment of LMMs for long video understanding.

To alleviate excessive memory consumption, we propose a progressive pooling strategy. As shown in Figure 3, we first uniformly divide the video sequence embeddings $F_v$ into multiple groups, with a division stride defined as $K$. We assume that each group represents an event. Considering the redundancy between consecutive frames in the same event, such as a static background, we retain only the first frame of each group at a higher spatial resolution. The remaining frames within each group are stored at a lower spatial resolution using a larger pooling stride. Specifically, the video sequence embeddings $F_v$ are divided into multiple groups, each containing $K$ frames, resulting in a total of $M = \frac{N}{K}$ groups.:

$$\{F_{v,i}\}_{i=1}^N \rightarrow \{\{F_{v,w,j}\}_{j=1}^K\}_{w=1}^M. \tag{6}$$

Table 1: Performance evaluation on VideoMME (Fu et al., 2024) benchmark. * indicates the results of reproduction.

| Methods | Frames | Short | Medium | Long | Overall |
|---|---|---|---|---|---|
| Qwen-VL-Chat-7B (Bai et al., 2023) | 4 | 46.9 | 38.7 | 37.8 | 41.1 |
| VideoLLaVA-7B (Lin et al., 2023) | 8 | 45.3 | 38.0 | 36.2 | 39.9 |
| VideoChat2-Mistral-7B (Li et al., 2024b) | 16 | 48.3 | 37.0 | 33.2 | 39.5 |
| VideoLLaMA2-7B (Cheng et al., 2024) | 16 | 56.0 | 45.4 | 42.1 | 47.9 |
| LLaVA-NeXT-Qwen2-7B (Liu et al., 2024b) | 32 | 58.0 | 47.0 | 43.4 | 49.5 |
| LLaVA-OneVision-7B* (Li et al., 2024a) | 32 | 69.3 | 55.1 | 49.7 | 58.2 |
| Chat-UniVi-V1.5-7B (Jin et al., 2024) | 64 | 45.7 | 40.3 | 35.8 | 40.6 |
| ST-LLM-7B (Liu et al., 2024d) | 64 | 45.7 | 36.8 | 31.3 | 37.9 |
| LongVA-7B (Zhang et al., 2024) | 128 | 61.1 | 50.4 | 46.2 | 52.6 |
| LongVILA-8B (Xue et al., 2024) | 256 | 61.8 | 49.7 | 39.7 | 50.5 |
| **Ours** | 256 | **72.7** | 58.2 | **52.9** | **61.3** |
|  | 512 | 71.9 | **58.7** | 51.3 | 60.6 |

In each group, the first frame $F_{v,w,1}$ is retained at high resolution:

$$F_{v,w,1}^{\text{high-res}} = \text{Pool}(F_{v,w,1}, \text{stride} = s_h). \tag{7}$$

The remaining frames are pooled at a lower resolution with a larger stride $s_l$ ($s_h < s_l$), resulting in:

$$\{F_{v,w,j}^{\text{low-res}} = \text{Pool}(F_{v,w,j}, \text{stride} = s_l)\}_{j=2}^{K}. \tag{8}$$

Finally, the processed frames are reassembled into a new video sequence embedding $F_v^{\text{new}}$:

$$F_v^{\text{new}} = \{\{F_{v,w,1}^{\text{high-res}}, F_{v,w,2}^{\text{low-res}}, \ldots, F_{v,w,K}^{\text{low-res}}\}_{w=1}^{M}\}, \tag{9}$$

where $\text{Pool}(\cdot, \text{stride})$ represents the pooling operation with the specified stride.

The progressive pooling strategy significantly reduces the number of visual tokens while preserving the integrity of spatial information.

# 4 EXPERIMENTS

## 4.1 EXPERIMENT SETTING

We evaluate the long video understanding capabilities of our method on three key benchmarks: VideoMME (Fu et al., 2024), MLVU (Zhou et al., 2024), and LongVideoBench (Wu et al., 2024).

**VideoMME** is a widely used benchmark for assessing the ability of LMMs to handle long videos in real-world scenarios. It divides the test set into three subsets based on video length: short videos ($<$ 2 minutes), medium-length videos (4 to 15 minutes), and long videos (30 to 60 minutes), with durations ranging from 11 seconds to 1 hour.

**MLVU** offers a diverse collection of video lengths, types, and evaluation tasks. It includes long video understanding tasks (TR: Topic Reasoning, AR: Anomaly Recognition), single-detail long video understanding tasks (NQA: Needle QA, ER: Ego Reasoning, PQA: Plot QA), and multi-detail long video understanding tasks (AO: Action Order, AC: Action Count). The benchmark includes videos of various types, such as movies, surveillance footage, egocentric videos, cartoons, and game videos, with lengths ranging from 3 minutes to over 2 hours.

**LongVideoBench** focuses on long-span understanding, particularly on referential reasoning problems that depend on long-frame inputs and cannot be resolved using only a single or sparse frames. It evaluates videos of varying lengths, including (8s, 15s], (15s, 60s], (180s, 600s], and (900s, 3600s].

## 4.2 IMPLEMENTATION DETAILS

To validate the effectiveness of our approach, we use the latest LMM, LLaVA-OneVision 7B, as the backbone and baseline model. This model employs a classic multimodal encoder-decoder architecture, consisting of a visual encoder (SigLIP (Zhai et al., 2023)), an LLM decoder (Qwen2), and

Table 2: The overall performances on MLVU (Zhou et al., 2024). Two input strategies are used by the LMMs in evaluation: Uniform Sampling, which evenly samples $N$ frames from the video; Frame Rate Sampling (N fps), which samples $N$ frames per second. † denotes proprietary models.

| Methods | Frames | Holistic | | Single Detail | | | Multi Detail | | M-Avg |
|---|---|---|---|---|---|---|---|---|---|
| | | TR | AR | NQA | ER | PQA | AO | AC | |
| GPT-4o† (OpenAI, 2024) | 0.5 fps | 87.4 | 74.5 | 64.8 | 57.1 | 65.1 | **56.7** | **46.3** | 64.6 |
| LLaMA-VID-7B (Li et al., 2023d) | 1 fps | 50.8 | 34.5 | 30.1 | 32.7 | 32.5 | 23.9 | 27.8 | 33.2 |
| LLaVA-1.6-7B (Liu et al., 2024b) | 16 | 60.6 | 41.0 | 43.1 | 38.4 | 41.0 | 25.5 | 25.7 | 39.3 |
| InternVL-1.5-7B Chen et al. (2024b) | 16 | 78.8 | 67.0 | 52.7 | 43.5 | 54.4 | 32.8 | 23.8 | 50.4 |
| LLaVA-OneVision-7B* (Li et al., 2024a) | 32 | **88.6** | 74.0 | 73.0 | 62.2 | 67.9 | 43.2 | 28.6 | 64.2 |
| TimeChat-7B (Ren et al., 2024) | 96 | 23.1 | 27.0 | 24.5 | 28.4 | 25.8 | 24.7 | 32.0 | 30.9 |
| LongVA-7B (Zhang et al., 2024) | 256 | 83.3 | 58.5 | 69.3 | 50.0 | 67.2 | 38.6 | 27.2 | 56.3 |
| MovieChat-7B (Song et al., 2024) | 2048 | 29.5 | 25.0 | 24.2 | 24.7 | 25.8 | 28.6 | 22.8 | 25.8 |
| **Ours** | 256 | 87.5 | 74.5 | **76.3** | 65.3 | 75.9 | 52.9 | 31.6 | 68.6 |
| | 512 | 87.1 | **76.5** | 75.5 | **65.3** | **76.1** | 52.5 | 37.4 | **69.1** |

Table 3: Performance evaluation on LongVideoBench (Wu et al., 2024) benchmark.

| Methods | Frames | Duration Group (s) | | | | Avg |
|---|---|---|---|---|---|---|
| | | (8, 15] | (15, 60] | (180, 600] | (900, 3600] | |
| LLaVA-1.5-13B (Liu et al., 2024a) | 8 | 49.0 | 51.1 | 41.8 | 39.6 | 43.4 |
| LLaVA-Next-Mistral-7B (Liu et al., 2024b) | 8 | 53.4 | 57.2 | 46.9 | 42.1 | 49.1 |
| VideoLLaVA-7B (Lin et al., 2023) | 8 | 43.1 | 44.6 | 36.4 | 34.4 | 39.1 |
| VideoChat2-7B (Li et al., 2024b) | 8 | 49.3 | 49.3 | 39.0 | 37.5 | 39.3 |
| LLaVA-Next-Video-34B (Liu et al., 2024b) | 8 | 57.6 | 61.6 | 48.7 | 45.9 | 50.5 |
| PLLaVA-34B (Xu et al., 2024) | 8 | 60.1 | 66.8 | 50.8 | 49.1 | 53.2 |
| LLaVA-OneVision-7B* (Li et al., 2024a) | 32 | 68.8 | **70.4** | 54.6 | 48.1 | 56.0 |
| LongVA-7B (Zhang et al., 2024) | 256 | 57.4 | 60.4 | 47.3 | 44.7 | 49.7 |
| **Ours** | 256 | **68.8** | 69.2 | 56.1 | 51.2 | 57.5 |
| | 512 | 66.1 | 67.4 | **58.5** | **52.1** | **58.0** |

a multimodal projection module (MLP). For each video frame, the visual encoder and multimodal projection module encode the frame into video sequence embeddings $F_v \in \mathbb{R}^{N \times 729 \times d}$. Through bilinear pooling with a stride of 2, this is reduced to $F_v \in \mathbb{R}^{N \times 196 \times d}$.

Following the default settings in YaRN, we set the hyperparameters $\alpha$ and $\beta$ (in Section 3.1) to 1 and 32, respectively. Previous research (Peng et al., 2024; Chen et al., 2023b; Ding et al., 2024) has shown that fine-tuning after interpolation enhances a model's ability to interpret scaled RoPE embeddings. Therefore, we compare the results of both tuning-free and fine-tuned approaches. Specifically, we randomly sample 10K instances from the allava instruction dataset (Chen et al., 2024a) and fine-tune the LLaVA-OneVision language decoder using LoRA (Hu et al., 2022), setting $lora\_r$ to 64 and $lora\_\alpha$ to 16. The learning rate is set to $1e-5$ with a batch size of 1. In our experiments, unless otherwise stated, we use the tuning-free model to present our results. The default parameters for the progressive pooling method are: division stride $K = 4$, high-resolution pooling stride $s_h = 2$, and low-resolution pooling stride $s_l = 8$.

## 4.3 QUATITATIVE RESULTS

**Results on VideoMME** Table 1 presents the results on the VideoMME benchmark. Compared to the baseline model, LLaVA-OneVision, our method shows consistent improvements across all intervals for short, medium, and long videos. Notably, for long videos, the accuracy improved by 3.2%. In comparison to the latest long video understanding models, our approach continues to achieve optimal performance. For instance, compared to LongVILA-8B, which was pre-trained on long video-text pairs, our method demonstrates an improvement of 10.8%. Crucially, our method achieves these gains without requiring any pre-training or fine-tuning on long video-text pairs.

Table 4: Performance evaluation of different context window extension methods on the VideoMME.

| | Frames | Short | Medium | Long | Overall |
|---|---|---|---|---|---|
| LLaVA-OneVision-7B | 256 | 64.9 | 53.3 | 50.4 | 56.2 |
| LLaVA-OneVision-7B + YaRN | 256 | 67.6 | 56.3 | 51.7 | 58.5 |
| Ours (Tuning-free) w/o progressive pooling | 256 | 71.6 | 59.1 | 52.2 | 61.0 |
| Ours (Fine-tuning) w/o progressive pooling | 256 | **71.9** | **60.2** | **53.2** | **61.8** |

Table 5: Ablation studies on the VideoMME benchmark, where all videos are uniformly sampled to 256 frames. Specifically, $s_h$ represents the high-resolution pooling stride for the first frame of each group; $s_l$ indicates the low-resolution pooling stride for the remaining frames within each group; and $K$ denotes the grouping stride, which refers to the number of frames within each group.

| $(s_h, s_l), K$ | Memory (GB) | Short | Medium | Long | Overall |
|---|---|---|---|---|---|
| (2, 2), 0 | 73 | 71.6 | **59.1** | 52.2 | 61.0 |
| (4, 4), 0 | 37 | 70.8 | 59.0 | 51.2 | 60.3 |
| (8, 8), 0 | 29 | 68.1 | 56.2 | 49.7 | 58.0 |
| (2, 4), 4 | 45 | 72.4 | 58.3 | 51.3 | 60.7 |
| (2, 8), 4 | 40 | **72.7** | 58.2 | **52.9** | **61.3** |
| (2, 4), 8 | 41 | 70.1 | 57.6 | 50.8 | 59.5 |
| (2, 8), 8 | 35 | 69.7 | 56.4 | 51.4 | 59.2 |
| (2, 4), 16 | 40 | 68.6 | 57.4 | 51.4 | 59.1 |
| (2, 8), 16 | 31 | 70.3 | 56.3 | 50.7 | 59.1 |

**Results on MLVU and LongVideoBench** MLVU and LongVideoBench are two benchmarks specifically designed to evaluate long video understanding tasks. Table 2 presents the results on MLVU, where our method significantly outperforms all comparison models, even surpassing GPT-4o. Table 3 provides the results on LongVideoBench, where test samples are categorized into various duration intervals to highlight different models' performance in long video comprehension. Our method shows a slight performance drop in the intervals (8, 15] and (15, 60] when sampling 512 frames compared to the baseline LLaVA-OneVision. This performance drop in shorter intervals can be attributed to the fact that dense frame sampling results in excessively long input sequences for shorter videos, which leads to attention distraction and degrades model performance. Using different frame sampling strategies for videos of varying durations can alleviate this issue.

## 4.4 ABLATION STUDIES

To validate the effectiveness of the proposed module, we conducted experiments on VideoMME, focusing on visual context window extension and progressive pooling strategies.

**Visual Context Window Extension** Table 4 presents the comparative results under the scenario of uniformly sampling 256 frames, including direct extrapolation, YaRN interpolation, and our method. It is noteworthy that all results in the table did not utilize the progressive pooling strategy. The results indicate that using YaRN interpolation improves model performance, confirming the effectiveness of positional interpolation. Our method, which applies interpolation on the visual context window, achieves a significant performance enhancement compared to YaRN. Additionally, we fine-tuned the model using 10K image-text pairs after interpolation, further improving model performance. This aligns with the conclusions drawn from context window extension methods in LLMs.

**Progressive Pooling** Table 5 presents the comparative results of different pooling strategies and progressive pooling parameters on VideoMME. It is important to note that all experiments in the table utilized visual context window extension. The upper half of the table displays the results of uniform pooling with pooling strides of 2 (the default pooling strategy of the baseline model), 4, and 8. It is evident that as the pooling stride increases, memory consumption decreases gradually, but performance declines progressively. The lower half of the table shows the results of our proposed progressive pooling strategy. We conducted experiments with varying pooling strides and grouping strides, comparing performance under different parameters. The results indicate that the optimal performance occurs at $s_h = 2$, $s_l = 8$, and $K = 4$. In this setting, compared to the baseline method

**LLaVA-OneVision-7B 32 frames**

> …The scene transitions to an outdoor track and field stadium filled with spectators, where athletes are preparing for their events. …. The javelin thrower is seen kneeling on the track, celebrating their performance…

**LLaVA-OneVision-7B 256 frames (extrapolation)**

> The video begins with a close-up of the IAAF World Championships Moscow 2013 logo, followed by a series of images showing an athlete in a red and black uniform holding a javelin. The athlete is seen standing on a track field with a large stadium filled with spectators in the background. The athlete is also shown celebrating, raising their arms and waving the flag of Germany….

**Ours 256 frames**

> … It shows a female athlete dressed in a red and black uniform with the word "GERMANY" printed across her chest, indicating her nationality. She is seen running along the track with a javelin in hand, preparing for her throw. … The final segment of the video highlights the athlete\'s victory celebration. She is shown holding the German flag, waving it with enthusiasm, and interacting with photographers and officials….

Figure 4: Qualitative results from different methods demonstrate that our approach exhibits accurate and detailed video captioning capabilities. Red indicates incorrect content, while blue represents the corresponding correct description.

(with a uniform pooling stride of 2), our approach reduces memory usage by approximately 45% while achieving superior performance. This is because shorter sequence lengths mitigate the issue of attention distraction. Additionally, we found that the pooling stride has a smaller impact on the model, while the grouping stride has a significant effect. This may be due to larger grouping strides leading to greater intra-group scene variation, resulting in a loss of spatial information.

### 4.5 QUALITATIVE RESULTS

Figure 4 illustrates the qualitative results of our method in video captioning. It is evident that LLaVA-OneVision-7B generates incorrect descriptions when the default input is set to 32 frames. When directly extrapolated to 256 frames, the model appears to forget information from the middle section of the video, only describing the beginning and the end. In contrast, our method generates accurate and detailed descriptions for the input video when 256 frames are provided. Appendix A.4 provides the complete video descriptions.

### 4.6 CONCLUDING REMARKS

In this paper, we address the long video understanding issue from the perspective of context windows, effectively avoiding the resource consumption associated with training from scratch. By redefining the effective context window of LMMs into visual and language context windows, we propose the visual context window extension. This approach allows LMMs trained on short videos to be applied to long video understanding tasks without fine-tuning. Additionally, we introduce a progressive pooling strategy to mitigate memory consumption issues caused by long sequences. In a 256-frame setting, this strategy reduces memory usage by approximately 45% without introducing any performance loss. We hope this work will advance research in long video understanding and provide insights for the design of future long video understanding models.

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

## A APPENDIX

### A.1 VISUAL NEEDLE-IN-A-HAYSTACK

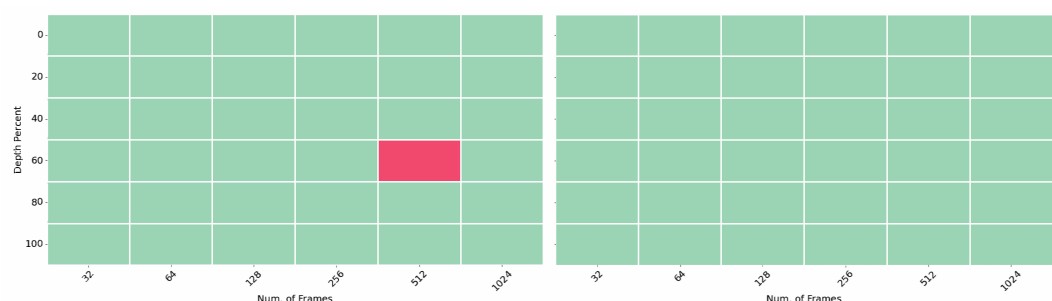

Figure 5: Visualization of the Needle in the Long Video Haystack Experiment, where green represents correct answers, while red indicates incorrect answers. Left: progressive pooling parameters are set to $s_h = 2$, $s_l = 8$, $K = 4$. Right: progressive pooling parameters are set to $s_h = 2$, $s_l = 4$, $K = 4$. Our method enables LMMs, pre-trained on short videos (32 frames), to be extended to 1024 frames without requiring fine-tuning.

As shown in Figure 5, we utilize V-NIAH (Zhang et al., 2024) to measure the model's long-context capabilities. Probes are inserted at different positions within the video, and a question-answering task is conducted; a response is considered correct only when it matches the answer (indicated in green), otherwise, it is deemed incorrect (indicated in red). It is evident that our method demonstrates outstanding performance across different progressive pooling parameters, effectively extending the model's visual context window to 1024 frames without requiring fine-tuning.

### A.2 ROTARY POSITION EMBEDDINGS

Rotational Position Embeddings (RoPE) (Su et al., 2024) introduce a rotation matrix and meanwhile incorporates the explicit relative position dependency in self-attention formulation, enabling the model to capture the relative positional relationships between words, thereby enhancing its performance in processing sequential data.

Given a sequence $\mathbf{S} = \{\mathbf{w}_i\}_{i=1}^N$, where $N$ represents the sequence length and $w_i$ represents the $i$-th word. Its corresponding word embeddings are $\mathbf{E} = \{\mathbf{x}_i\}_{i=1}^N$, where $\mathbf{x}_i$ is the embedding of the $i$-th word. Before calculating attention, it is necessary to incorporate positional information into the word embeddings and transform them into the query vectors and the key vectors.

$$\mathbf{q}_m = f_q(\mathbf{x}_m, m) \in \mathbb{R}^d, \quad \mathbf{k}_n = f_k(\mathbf{x}_n, n) \in \mathbb{R}^d, \tag{10}$$

where $m$ and $n$ represent different positions, respectively. Next, attention is computed using the query and key vectors.

$$\text{softmax}\left(\frac{\mathbf{q}_m^T \mathbf{k}_n}{\sqrt{d}}\right), \tag{11}$$

where $\mathbf{q}_m, \mathbf{k}_n$ are considered as column vectors so that $\mathbf{q}_m^T \mathbf{k}_n$ is simply the Euclidean inner product.

To incorporate relative positional information, we express the inner product between the query and key vectors as a function, denoted as $g(\cdot)$.

$$\langle f_q(\mathbf{x}_m, m), f_k(\mathbf{x}_n, n) \rangle = g(\mathbf{x}_m, \mathbf{x}_n, m - n). \tag{12}$$

For the function $g(\cdot)$, it is evident that the inner product encodes positional information only in a relative form (i.e., $m - n$).

The next goal is to find an appropriate function $g(\cdot)$ that conforms to the aforementioned relation. Specifically, we first represent the query and key vectors in complex form. The representations of the query and key vectors are as follows:

$$f_q(\mathbf{x}_m, m) = e^{im\Theta}(\mathbf{W}_q \mathbf{x}_m), \quad f_k(\mathbf{x}_n, n) = e^{in\Theta}(\mathbf{W}_k \mathbf{x}_n). \tag{13}$$

For the sake of clarity and ease of understanding in the subsequent formulas, where $\mathrm{i}^2 = -1$ is the imaginary unit and $\Theta = \mathrm{diag}\left(\theta_j = b^{-2j/d}, j \in [1, 2, \ldots, d/2]\right)$ is the diagonal matrix. RoPE associates each (complex-valued) hidden neuron with a distinct frequency $\theta_j$. The benefit of this approach is that the dot product between the query and key vectors depends only on the relative distance $m - n$. This process is represented by the following formula:

$$
\begin{aligned}
& \langle f_q\left(\mathbf{x}_m, m\right), f_k\left(\mathbf{x}_n, n\right)\rangle \\
& = \left\langle e^{im\Theta}\left(\mathbf{W}_q \mathbf{x}_m\right), e^{in\Theta}\left(\mathbf{W}_k \mathbf{x}_n\right)\right\rangle \\
& = \mathrm{Re}\left(e^{i\Theta(m-n)} \mathbf{x}_m^* \mathbf{W}_q^* \mathbf{W}_k \mathbf{x}_n\right) \\
& = g\left(\mathbf{x}_m, \mathbf{x}_n, m - n\right),
\end{aligned}
\tag{14}
$$

where $\mathrm{Re}(\cdot)$ is the real part of a complex number and $(\cdot)^*$ represents the conjugate complex number of $(\cdot)$.

According to Euler's formula,

$$
e^{i(m-n)\Theta} = \cos((m - n)\Theta) + i\sin((m - n)\Theta).
\tag{15}
$$

In real coordinates, RoPE can be expressed using the following function:

$$
f_q\left(\mathbf{x}_m, m\right) = \mathcal{R}_m\left(\mathbf{W}_q \mathbf{x}_m\right) =
$$
$$
\begin{pmatrix}
\cos m\theta_1 & -\sin m\theta_1 & \cdots & 0 & 0 \\
\sin m\theta_1 & \cos m\theta_1 & \cdots & 0 & 0 \\
0 & 0 & \cdots & 0 & 0 \\
0 & 0 & \cdots & 0 & 0 \\
0 & 0 & \cdots & \cos m\theta_{d/2} & -\sin m\theta_{d/2} \\
0 & 0 & \cdots & \sin m\theta_{d/2} & \cos m\theta_{d/2}
\end{pmatrix} \mathbf{W}_q \mathbf{x}_m.
\tag{16}
$$

Therefore, when the word embedding $\mathbf{x}_m$ at position $m$ is multiplied by matrix $\mathcal{R}_m$, and the word embedding $\mathbf{x}_n$ at position $n$ is also multiplied by matrix $\mathcal{R}_n$, resulting in the transformed query and key vectors, the attention weights will inherently include the relative positional information. This is because the following identity holds:

$$
\begin{aligned}
& \left(\mathcal{R}_m \mathbf{W}_q \mathbf{x}_m\right)^\top \left(\mathcal{R}_n \mathbf{W}_k \mathbf{x}_n\right) \\
& = \left(\mathbf{W}_q \mathbf{x}_m\right) \mathcal{R}_m^\top \mathcal{R}_n \left(\mathbf{W}_k \mathbf{x}_n\right) \\
& = \left(\mathbf{W}_q \mathbf{x}_m\right)^\top \mathcal{R}_{n-m} \left(\mathbf{W}_k \mathbf{x}_n\right).
\end{aligned}
\tag{17}
$$

### A.3 VISUAL CONTEXT WINDOW EXTENSION

In this section, we provide a more detailed derivation of the visual context window extension based on YaRN.

Unlike the context window extension methods used in LLMs, we first define the visual context window ($L_{\text{train}}^v$), and the extended context window ($L_{\text{test}}^v$), with the scale factor $s$ representing the ratio between the two:

$$
s = \frac{L_{\text{test}}^v}{L_{\text{train}}^v}.
\tag{18}
$$

Based on the derivation of RoPE, the inner product between the query and key vectors can be expressed in complex form as follows:

$$
\left(\mathcal{R}_m \boldsymbol{q}\right)^\top \left(\mathcal{R}_n \boldsymbol{k}\right) = \mathrm{Re}\left[\sum_{i=0}^{d/2-1} \boldsymbol{q}_{[2i:2i+1]} \boldsymbol{k}_{[2i:2i+1]}^* e^{i(m-n)\theta_i}\right]
\tag{19}
$$

where $\boldsymbol{q} = \mathbf{W}_q \mathbf{x}_m$ and $\boldsymbol{k} = \mathbf{W}_k \mathbf{x}_n$. According to Euler's formula, $e^{i(m-n)\theta_i}$ can be represented as a point on the unit circle, where $m - n$ controls the angle on the circle. Therefore, we define $\lambda_d$ as the wavelength of the RoPE embedding in the $d$-th hidden dimension.

$$
\lambda_i = \frac{2\pi}{\theta_i} = 2\pi b^{\frac{2i}{d}}.
\tag{20}
$$

The wavelength describes the token length required for the RoPE embedding to complete a full rotation $(2\pi)$ in dimension $d$.

Next, we define $r$, which represents the ratio between the original context size and the wavelength.

$$r(i) = \frac{L}{\lambda_i}. \tag{21}$$

This ratio determines which positional dimensions require interpolation. Following YaRN, we introduces two hyperparameters to control the boundaries of the interpolation strategy.

$$\theta_i^{\text{new}} = \left[\gamma_i + (1 - \gamma_i)\frac{1}{s}\right]\theta_i, \quad \gamma_i = \begin{cases} 1, & r_i > \beta \\ 0, & r_i < \alpha \\ \frac{r_i - \alpha}{\beta - \alpha}, & \text{otherwise,} \end{cases} \tag{22}$$

When $r_i < \alpha$, linear interpolation is applied proportionally based on $s$. When $r_i > \beta$, no interpolation is applied. Otherwise, a linear interpolation transition is applied between the above two cases.

## A.4 LONG VIDEO CAPTION

Figure 6 presents the complete generation of qualitative results for video captioning. It is evident that the descriptions produced by our method are more accurate.

## A.5 MORE VISUALIZATION

Figure 7 illustrates the visualization of visual and language embeddings from different layers of the LLM decoder after applying the visual context window extension. It can be observed that even after the application of the context window extension, the visual and language embeddings still exhibit two clustering centers. This is because the visual context window extension strategy is not designed to address the differences between modalities. Instead, its primary goal is to scale the positional embeddings of visual tokens to fit within the effective visual context window, enabling the model to process longer visual sequences without additional training. By doing so, the method cost-effectively adapts LMM trained on short videos to handle long video sequences.

**LLaVA-OneVision-7B 32 frames**

The video begins with a close-up of the IAAF World Championships Moscow 2013 logo, featuring a dynamic athlete in mid-air against a vibrant background of red, orange, and white colors. The scene transitions to an outdoor track and field stadium filled with spectators, where athletes are preparing for their events. A javelin thrower, dressed in a red and black uniform with yellow shoes, is seen holding a javelin and running towards the throwing area. The background includes various sponsor banners such as 'SEIKO,' 'Canon,' and 'adidas.' The javelin thrower then releases the javelin into the air, with the camera capturing the motion from different angles.\n\nThe video continues with a close-up of the javelin thrower in mid-action, running on the track with the javelin in hand. The background shows the crowded stadium with spectators and officials. The scene then shifts to a wider view of the stadium, revealing the javelin thrower in the foreground with the crowd in the background. The javelin thrower is seen releasing the javelin into the air, with the camera capturing the motion from different angles. The video concludes with the javelin thrower kneeling on the track, celebrating their performance.\n\nThe final segment of the video features a close-up of the javelin thrower in mid-action, running on the track with the javelin in hand. The background shows the crowded stadium with spectators and officials. The javelin thrower is seen releasing the javelin into the air, with the camera capturing the motion from different angles. The scene then shifts to a wider view of the stadium, revealing the javelin thrower in the foreground with the crowd in the background. The javelin thrower is seen kneeling on the track, celebrating their performance. The video concludes with the javelin thrower standing up, holding a German flag, and waving it triumphantly. The background includes the stadium's seating and various event banners. The text 'IAAF JAVELIN THROW WOMEN' and 'Moscow 2013' appears at the bottom of the screen, along with the athlete's name and results.

**LLaVA-OneVision-7B 256 frames (extrapolation)**

The video begins with a close-up of the IAAF World Championships Moscow 2013 logo, followed by a series of images showing an athlete in a red and black uniform holding a javelin. The athlete is seen standing on a track field with a large stadium filled with spectators in the background. The athlete is also shown celebrating, raising their arms and waving the flag of Germany.\n\nThe video then transitions to a wider shot of the stadium, showing the athlete walking across the field while still holding the flag. The stadium is filled with spectators, and there are banners and flags visible in the background. The athlete continues to celebrate, waving the flag and smiling.\n\nThe video concludes with a close-up of the athlete walking across the field, still holding the flag. The athlete is wearing a red and black uniform with the name "TOYOTA" and the number "431" on it. The athlete is also wearing black shorts and yellow shoes. The athlete is walking towards the camera, waving the flag and smiling. The background shows the stadium filled with spectators, and there are banners and flags visible in the background. The athlete continues to celebrate, waving the flag and smiling.

**Ours 256 frames**

The video begins with a series of images displaying the logo for the IAAF World Championships Moscow 2013. The logo features a stylized figure in motion, set against a backdrop of vibrant colors including red, white, and blue, symbolizing speed and athleticism.\n\nFollowing this, the video transitions to footage from an athletics event, specifically focusing on javelin throw. It shows a female athlete dressed in a red and black uniform with the word "GERMANY" printed across her chest, indicating her nationality. She is seen running along the track with a javelin in hand, preparing for her throw. The background reveals a crowded stadium filled with spectators, and various advertisements such as "SEIKO," "Canon," and "adidas" are visible around the track area.\n\nThe athlete then executes her javelin throw, running up to gain momentum before releasing the javelin into the air. The camera captures the javelin\'s flight path, showing it soaring through the air towards the field. The stadium is bustling with activity, and officials can be seen observing the event closely.\n\nThe video continues to show the athlete in mid-action, having just released the javelin. The camera follows the javelin\'s trajectory, which is captured in slow motion, emphasizing its speed and distance. The athlete is seen celebrating her throw, raising her arms in triumph, and the crowd in the stands appears engaged and excited.\n\nThe final segment of the video highlights the athlete\'s victory celebration. She is shown holding the German flag, waving it with enthusiasm, and interacting with photographers and officials. The scoreboard overlay indicates that she has won the event with a score of 96.58 meters, marked as "GOLD." The athlete is visibly elated, smiling broadly and embracing the moment of triumph. The background remains consistent with the previous scenes, featuring the large stadium filled with spectators and the event\'s branding prominently displayed.\n\nIn summary, the video showcases the IAAF World Championships Moscow 2013, focusing on the javelin throw event. It captures the athlete\'s preparation, execution, and celebration of her winning throw, highlighting the excitement and competitive spirit of the championship.

Figure 6: An example of generating long video captions using different methods. Compared to 32-frame and 256-frame extrapolations, our approach exhibits greater detail and accuracy.

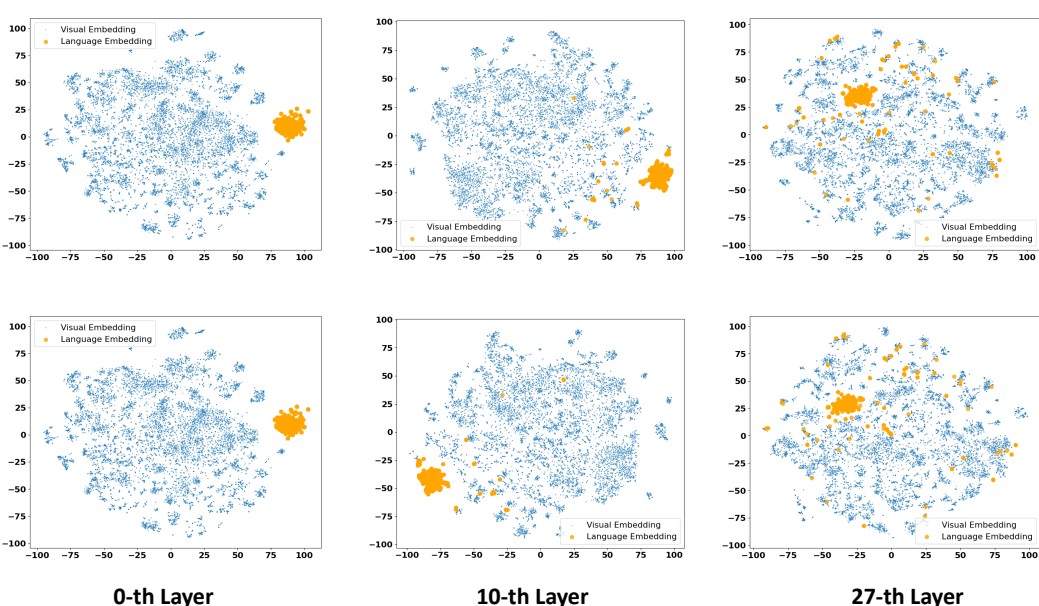

**0-th Layer**  **10-th Layer**  **27-th Layer**

Figure 7: Top: Visualization of visual and language embeddings from different layers of LLM decoder without the visual context window extension strategy; Down: Visualization of visual and language embeddings from different layers of LLM decoder with the visual context window extension strategy.

