# OpenReview forum: "Visual Context Window Extension: A New Perspective for Long Video Understanding"
_ICLR.cc/2025/Conference — Submitted to ICLR 2025_

### Official Review · Reviewer_eLe1 · 2024-10-17

**Soundness:** 3
**Presentation:** 3
**Contribution:** 3
**Rating:** 6
**Confidence:** 5

**Summary:**

This paper proposes visual context window extension of MLLMs for long video understanding, allowing them to process longer video sequences without retraining on long video datasets. Additionally, the paper uniformly pooling several frame tokens for memory concern. The method is evaluated across several benchmarks, showing competitive performance, surpassing GPT-4o on the MLVU benchmark.

**Strengths:**

+ The method adapt YaRN, originally designed for extending language context windows, to handle visual context windows separately.
+ The performance is impressive on several long-video benchmarks such as VideoMME, MLVU and LongVideoBench
+ The finetuning process does not require long video datasets.

**Weaknesses:**

+ Several theoretical insights are missing. Specifically, it is unclear why a uniformly sampled group is assumed to represent an event, and the rationale behind choosing to maintain the first frame at a high resolution is not explained. For further details, refer to questions 1, 2, 3.
+ The contribution is limited in terms of progressive pooling. Firstly, there is no progressive process. The method simply maintains the first frame at high resolution while spatially pooling the subsequent frames to a lower resolution.
+ The proposed strategy seems hard to handle the scenarios when the input consists of interleaved visual and text embeddings.

**Questions:**

1. In Figure 1(b), the diagram illustrates the modality gap between visual and language embeddings, which leads to the visual context window extending beyond its range, resulting in a decrease in performance. Could you clarify how extending the visual context helps to narrow this gap?
2. How does the clustering visualization of visual and language embeddings change after implementing the proposed window extension strategy?
3. In line 100, the author states that they "believe the first frame preserves rich information" and in line 264, they "assume each uniform group represents an event." These statements do not seem technically sound, and the author does not provide substantial evidence to support them. Given that a video contains dynamic content, a uniform sampling strategy might not effectively separate distinct events. Could you explain why the first frame is considered to preserve rich information?
4. Does this visual/text context extension support the interleaving of visual and text features?
5. How does the LLaVA-OneVision model perform when utilizing the full context length of Qwen2, instead of uniformly sampling 32 frames on MLVU and LongVideoBench?

---

> ### Author Response · Authors · 2024-11-22
>
> Thank you so much for the thoughtful questions and suggestions. We hope that our response below will address your concerns.
>
> **Q1**: …. Could you clarify how extending the visual context helps to narrow this gap?
>
> **R1**: Thank you for your insightful comment. In fact, extending the visual context window does not directly narrow the modality gap between visual and language embeddings. Instead, the primary goal of this approach is to scale the positional embeddings of visual tokens into an effective visual context window in a training-free manner. This enables the extension of LMM trained on short videos to handle long videos, without requiring additional training to align the two modalities.
>
> Recent works [1] have attempted to align visual and language embeddings, but these approaches typically require extensive retraining and do not specifically address the challenges of long video understanding.
>
> [1] Jiang T, Song M, Zhang Z, et al. E5-v: Universal embeddings with multimodal large language models. arXiv preprint arXiv:2407.12580, 2024.
>
> **Q2**: How does the clustering visualization of visual and language embeddings change after implementing the proposed window extension strategy?
>
> **R2**: Following your advice, we further visualized the clustering of visual and language embeddings at different layers of the LLM decoder after applying the visual context window extension. The results are presented in **Figure 7 of Appendix A.5**. As shown in the visualization, the visual and language embeddings still exhibit two distinct cluster centers after applying the context window extension.
>
> This observation further supports our response to Question 1. The visual context window extension is not designed to address the modality gap between visual and language embeddings. Instead, its primary purpose is to scale the positional embeddings of visual tokens into an effective visual context window in a training-free manner. This allows LMM trained on short videos to be extended to long videos at a low computational cost.
>
>
> **Q3**: …. Could you explain why the first frame is considered to preserve rich information?
>
> **R3**: We apologize for the confusion caused by our imprecise wording. The statement that the first frame "preserves rich information" is made in comparison to other frames within the same group, which are compressed using a larger pooling stride.
>
> We argue that continuously sampling frames from a video often results in adjacent frames containing highly similar background information, leading to redundancy in the visual content and increased memory consumption. To address this, we apply a smaller pooling stride to the first frame of each group, preserving its fine-grained visual details, while compressing the other frames within the group more aggressively to reduce memory usage.
>
> **Regarding the uniform grouping strategy**, we acknowledge that dynamic content in videos may not always align perfectly with uniform temporal divisions. However, compared to event-based segmentation, a uniform grouping strategy is simpler and avoids the need for additional event segmentation models, which are themselves challenging to design and computationally expensive. Furthermore, event-based segmentation often leads to non-uniform frame sampling across different events, introducing additional complexity in hyperparameter tuning.
>
> In summary, to ensure **simplicity and efficiency**, we adopt a uniform grouping strategy as an empirical design choice. This approach avoids the overhead of event segmentation while still enabling a reasonable balance between preserving critical information and reducing memory consumption.

---

> > ### Author Response · Authors · 2024-11-22
> >
> > **Q4**: Does this visual/text context extension support the interleaving of visual and text features?
> >
> > **R4**: Thank you for your insightful question. In theory, the proposed context extension method can be applied to tasks involving interleaved visual and text features. This would only require identifying the model's effective context window and scaling both visual and text features to fit within the same context window.
> >
> > However, we note that existing visual-text interleaving benchmarks [1][2] typically involve relatively short inputs, with the number of images rarely exceeding 15. This presents a challenge for quantitatively evaluating our method, as our approach is specifically designed to enhance the model's ability to process long sequences.
> >
> > In future work, we plan to explore long-form visual-text interleaving benchmarks to further extend the applicability of our method and provide a more comprehensive evaluation.
> >
> > [1] Li F, Zhang R, Zhang H, et al. Llava-next-interleave: Tackling multi-image, video, and 3d in large multimodal models. arXiv preprint arXiv:2407.07895, 2024.
> >
> > [2] Tian C, Zhu X, Xiong Y, et al. Mm-interleaved: Interleaved image-text generative modeling via multi-modal feature synchronizer. TMLR, 2024.
> >
> >
> > **Q5**: How does the LLaVA-OneVision model perform when utilizing the full context length of Qwen2, instead of uniformly sampling 32 frames on MLVU and LongVideoBench?
> >
> > **R5**: Thank you for your suggestion. We evaluated our method by fixing the sampling rate to 256 frames on both the MLVU and LongVideoBench benchmarks. The results are summarized in the table below:
> >
> > MLVU:
> > | **Method** | TR | AR | NQA | ER | PQA | AO | AC | M-Avg |
> > |---------------|---------------|--------------|--------------|--------------|--------------|--------------|--------------|--------------|
> > | **LLaVA-OneVision(256 frames)** | 88.3    | 74.5        | 66.5        | 59.7        | 66.2        | 41.7        | 34.5        | 62.7        |
> > | **Ours (256 frames)** | 87.5        | 74.5        | 76.3        | 65.3        | 75.9        | 52.9        | 31.6        | 68.6        |
> >
> > LongVideoBench:
> > | **Method** |  (8, 15] | (15, 60] | (180, 600] | (900, 3600] | Overall |
> > |---------------|--------------|--------------|--------------|--------------|--------------|
> > | **LLaVA-OneVision (256 frames)**   | 61.9       | 67.4        | 52.4        | 47.0        | 53.4        |
> > |  **Ours (256 frames)**  | 68.8        | 69.2       | 56.1        | 51.2        | 57.5        |
> >
> > As shown in the results, our method significantly improves the long video understanding capabilities of the LLaVA-OneVision model on both MLVU and LongVideoBench benchmarks.

---

> > > ### Comment · Area_Chair_5HeD · 2024-11-24
> > >
> > > Dear eLe1,
> > >
> > > Could you please take a careful look at the other reviews and author responses, and comment on whether your original rating stands? Thank you.
> > >
> > > Best, AC

---

> > > > ### Comment · Reviewer_eLe1 · 2024-11-25
> > > >
> > > > Thanks for the response. My concerns have been resolved after the rebuttal.
> > > >
> > > > + The visual context length extension is not designed to address the modality gap between visual and language embeddings, but help to take longer visual input.
> > > > + The model could not be easily evaluated on interleaving of visual and text features but it could be explored in future work.
> > > > + The model performs very well on 256 frames input compared with baselines.
> > > >
> > > > It will be better to integrate Q1 and Q3 in the revision to avoid any confusion. I hope the code can be released to contribute the community for long video understanding.
> > > >
> > > > Hence, I am willing to increase my rating to 6.

---

> > > > > ### Author Response · Authors · 2024-11-26
> > > > >
> > > > > Thank you very much for your tremendous effort and valuable feedback on our paper. Your comments have been instrumental in helping us refine and improve the quality of our work. We are delighted to hear that your concerns have been resolved after the rebuttal.
> > > > >
> > > > > In response to your suggestion, we have integrated the answers to Q1 and Q3 in the revised version of the paper (highlighted in blue on page 2) to ensure clarity and avoid any potential confusion for future readers.
> > > > >
> > > > > We are committed to contributing to the open-source community by recently releasing our code and model checkpoints to promote long video understanding.
> > > > >
> > > > > If you have any additional concerns or suggestions, please do not hesitate to reach out. We are more than happy to address them promptly.

---

> > > > > > ### Author Response · Authors · 2024-11-26
> > > > > >
> > > > > > Dear Reviewer #eLe1,
> > > > > >
> > > > > > Thank you very much for your tremendous effort and valuable feedback on our paper. To support the research community and facilitate progress in long video understanding tasks, we have **provided the code for our method in the supplementary materials**. We hope this will contribute to the development of open-source tools and inspire further research in this area.
> > > > > >
> > > > > > Sincerely, Authors of Paper #2062

---

### Official Review · Reviewer_rZut · 2024-11-01

**Soundness:** 3
**Presentation:** 4
**Contribution:** 3
**Rating:** 6
**Confidence:** 4

**Summary:**

The paper focuses on adapting LMMs for long videos understanding. The authors propose extending the visual context window to adapt LMMs for long videos. They introduce a pooling strategy to reduce memory usage by selectively lowering the resolution of frame embeddings. This approach improves performance on long video tasks, while cutting memory usage by 45% without losing accuracy.

**Strengths:**

1.  The motivation is clear. Figure (a) effectively illustrates the challenge of long video understanding due to the limited visual context window.

2.  The proposed visual context window extension significantly improves baseline performance without additional tuning.

3.  The paper is well-written and easy to follow.

**Weaknesses:**

Overall, the technical contribution appears limited, as it primarily involves modifications to YaRN.

Additionally, Table 3 shows that the proposed method performs worse on short videos.

**Questions:**

The authors fixed the input frames to 256/512. Could this approach be extended to handle more frames?

What are the results of fine-tuning on 256 frames and then extending to a higher number of frames?

Could the authors provide results on MVbench? I'm interested in whether this visual context extension impacts the model's ability to understand short videos.

---

> ### Author Response · Authors · 2024-11-21
>
> Thank you very much for your positive and constructive feedback! We hope that our response below will address your concerns.
>
> **Q1**: The authors fixed the input frames to 256/512. Could this approach be extended to handle more frames?
>
> **R1**: In theory, our approach can be extended to handle even longer sequences of frames. However, processing more than 512 frames introduces significant memory overhead. Therefore, we only tested longer sequences (1024 frames) on the **VISUAL NEEDLE-IN-A-HAYSTACK** benchmark. As shown in **Appendix A.1**, our model achieved a **100% retrieval accuracy** with 1024-frame inputs.
>
> **Q1**: What are the results of fine-tuning on 256 frames and then extending to a higher number of frames?
>
> **R2**: Due to the quadratic computational complexity of attention mechanisms and the associated memory consumption, fine-tuning on 256 frames is extremely challenging. The input sequence length exceeds 50K tokens, which is computationally prohibitive, even in large-scale LLM training.
> As shown in **Table 1**, LongVILA [1] optimizes attention computation and trains on long videos with input lengths exceeding 256 frames, utilizing 256 A100 GPUs. In contrast, our proposed visual context window extension allows a model trained on only 32 frames to extend to 512 frames or even longer **without additional training**, while achieving superior performance.
>
> Our method is specifically designed to avoid such exorbitant training costs.
>
> [1] Xue F, Chen Y, Li D, et al. Longvila: Scaling long-context visual language models for long videos. arXiv preprint arXiv:2408.10188, 2024.
>
> **Q3**: Could the authors provide results on MVbench? …
>
> **R3**: In **Table 3** of the LongVideoBench evaluation, we observed that for short videos, sampling 512 frames led to a performance drop. This is because sampling 512 frames for 8-15 second short videos introduces significant visual redundancy. A straightforward solution to mitigate this issue is using **fps-based sampling** instead of fixed-parameter sampling. For LongVideoBench, we sampled frames at 1 fps, with a maximum of 512 frames.
> The results in the table below demonstrate that our method achieves significant performance improvements on short videos:
> |Method | (8, 15] | (15, 60] | (180, 600] | (900, 3600] | Overall |
> |----------|----------|----------|----------|----------|----------|
> | 512 frames| 66.1     | 67.4     | **58.5**     | 52.1     | 58.0     |
> | 1 fps |**70.4** | **73.3** | 57.8 | **52.8** | **59.4** |
>
> Additionally, following your suggestion, we evaluated our method on **MVBench**, and the results are summarized below. Results marked with * are reproduced using the same configuration:
>
> | Frames  | LLaVA-onevision| Ours (Tuning-Free) | Ours (Fine-Tuning) |
> |---------|-------------|-------------|-------------|
> | 32    | 56.2* | - | -      |
> | 64    | 56.0 | 56.4 | 56.5 |
> | 128 | 54.4 | 56.1 | 56.4 |
>
> For short videos, **dense sampling** (e.g., 128 frames) results in performance degradation due to increased redundancy. However, when fixed to 64 frames, our method still delivers performance improvements.

---

> > ### Comment · Area_Chair_5HeD · 2024-11-24
> >
> > Dear rZut,
> >
> > Could you please take a careful look at the other reviews and author responses, and comment on whether your original rating stands? Thank you.
> >
> > Best, AC

---

> ### Comment · Reviewer_rZut · 2024-11-25
>
> Thank you for your response. Some concerns have been addressed. I'd like to keep my rating.

---

> > ### Author Response · Authors · 2024-11-26
> >
> > Dear Reviewer #rZut,
> >
> > Thank you very much for your tremendous effort and valuable feedback on our paper.
> >
> > To support the research community and facilitate progress in long video understanding tasks, we have **provided the code for our method in the supplementary materials**. We hope this will contribute to the development of open-source tools and inspire further research in this area.
> >
> > If you have any other concerns or questions, please do not hesitate to let us know. We would be happy to address them promptly.
> >
> > Sincerely, Authors of Paper #2062

---

### Official Review · Reviewer_yZzs · 2024-11-02

**Soundness:** 3
**Presentation:** 3
**Contribution:** 3
**Rating:** 5
**Confidence:** 5

**Summary:**

The paper introduces a novel approach called Visual Context Window Extension for enhancing long video understanding using Large Multimodal Models (LMMs). It addresses the challenge by extending the visual context window without the need for retraining on long video datasets, leveraging the differences between visual and language modalities. The paper also proposes a progressive pooling strategy to reduce memory consumption, leading to improved performance and efficiency in long video understanding tasks.

**Strengths:**

This paper demonstrates a significant strength in addressing the limitations of Large Multimodal Models (LMMs) in long video understanding by proposing the Visual Context Window Extension, which leverages the inherent differences between visual and language modalities.
Another strength is the introduction of a progressive pooling strategy that effectively reduces memory consumption by approximately 45% without sacrificing performance, making the approach more feasible for practical applications.
 Lastly, the paper's approach shows consistent improvements in performance as the number of video frames increases, outperforming other models on benchmarks like MLVU, which is a substantial contribution to the field of long video understanding.

**Weaknesses:**

1. The progressive pooling strategy may also result in some loss of information.

2. The paper lacks a detailed overall framework diagram to showcase the entire process.

3. The paper may not be entirely fair in comparison with methods that use fewer frames, as a higher number of frames inherently contains more information.

**Questions:**

1. The progressive pooling strategy may also result in some loss of information.

2. The paper lacks a detailed overall framework diagram to showcase the entire process.

3. The paper may not be entirely fair in comparison with methods that use fewer frames, as a higher number of frames inherently contains more information.

---

> ### Author Response · Authors · 2024-11-21
>
> Thank you so much for your suggestion. We hope that our response below will address your concerns.
>
> **Q1**: The progressive pooling strategy may also result in some loss of information.
>
> **R1**: We reported the results of the ablation study on the progressive pooling strategy in **Table 5**. Under the (2,8),4 pooling strategy, our method achieved even better performance compared to the model without the progressive pooling strategy, while also reducing memory consumption by approximately **45%**.
>
> This improvement is due to the fact that continuous sampling of video frames can lead to similar backgrounds between adjacent frames, which generate redundant visual tokens. This redundancy results in longer input sequences, which can cause **attention distraction** and ultimately degrade model performance.
>
> The proposed **progressive pooling strategy** not only reduces memory consumption but also further improves the model's performance by mitigating the impact of redundant information.
>
> **Q2**: The paper lacks a detailed overall framework diagram to showcase the entire process.
>
> **R2**: In this paper, we proposed two key innovations: the **Visual Context Window Extension** and the **Progressive Pooling Strategy**. The former does not involve modifications to the model architecture, as it primarily optimizes the position embedding computation method for long-video understanding tasks, making it difficult to directly reflect in the model structure. As for the latter, the detailed processing pipeline of the progressive pooling strategy is visualized in **Figure 3**.
>
>
> **Q3**: The paper may not be entirely fair in comparison with methods that use fewer frames, as a higher number of frames inherently contains more information.
>
> **R3**: Our comparison includes both short-video LMMs, which support a small number of input frames (e.g., 4, 8, 16, 32, 64), and long-video LMMs, which support a higher number of frames (e.g., 128, 256, and 2048). Considering that the backbone of our method is a short-video LMM (LLaVA-OneVision), we included short-video LMMs in the comparison to establish a comprehensive evaluation benchmark.
>
> Moreover, as shown in **Figure 1 (a)** and **Table 4**, increasing the number of input frames for short-video LMMs can actually lead to a performance decline.

---

> > ### Comment · Area_Chair_5HeD · 2024-11-24
> >
> > Dear yZzs,
> >
> > Could you please take a careful look at the other reviews and author responses, and comment on whether your original rating stands? Thank you.
> >
> > Best, AC

---

> > ### Author Response · Authors · 2024-11-26
> >
> > Dear Reviewer #yZzs,
> >
> > Thank you very much for your tremendous effort and valuable feedback on our paper. Your comments are crucial for improving the quality of our work. In our latest response, we have carefully addressed your concerns and questions. If you have any other concerns or questions, please do not hesitate to let us know. We would be happy to address them promptly.
> >
> > Furthermore, to support the research community and facilitate progress in long video understanding tasks, we have **provided the code for our method in the supplementary materials**. We hope this will contribute to the development of open-source tools and inspire further research in this area.
> >
> > Sincerely, Authors of Paper #2062

---

> > > ### Author Response · Authors · 2024-12-01
> > >
> > > **In response to “The progressive pooling strategy may also result in some loss of information”.**
> > >
> > > Thank you for your insightful comment regarding the potential information loss caused by the progressive pooling strategy. We would like to provide a detailed explanation of the motivation behind this strategy and how it is designed to mitigate such concerns.
> > >
> > > The primary motivation for introducing the progressive pooling strategy in our work is to **reduce memory consumption** when processing long video inputs. It is true that directly applying pooling to video frames may lead to information loss. However, our **progressive pooling strategy** is specifically designed to minimize this issue by leveraging the inherent redundancy in video content.
> > >
> > > ### Key Design of Progressive Pooling:
> > > 1. **Grouping of Frames**:
> > >    Instead of treating all frames equally, we divide the video into **groups of frames**. This grouping allows us to handle frames within each group differently, balancing memory efficiency and information retention.
> > >
> > > 2. **Redundancy in Adjacent Frames**:
> > >    In most videos, adjacent frames often contain highly similar background information, leading to redundancy in the visual content. Directly sampling all frames without distinction results in unnecessary memory usage. To address this, we argue that not all frames require the same level of detail.
> > >
> > > 3. **Fine-Grained Details for Frames**:
> > >    Within each group, we apply a **smaller pooling stride** to the **first frame**, preserving its fine-grained visual details. For the remaining frames in the group, we apply a **larger pooling stride**, compressing them more aggressively to reduce memory consumption.
> > >
> > > ### Ablation Study Results:
> > > To validate the effectiveness of the progressive pooling strategy, we conducted an ablation study comparing different pooling parameters on the VideoMME benchmark. The results are summarized in the table below:
> > >
> > > | **($s_h$, $s_l$), $K$** | **Memory (GB)** | **Short** | **Medium** | **Long** | **Overall** |
> > > |------------------|-----------------|-----------|------------|-----------|-------------|
> > > | (2, 2), 0       | 73              | 71.6      | **59.1**       | 52.2      | 61.0        |
> > > | (4, 4), 0       | 37              | 70.8      | 59.0       | 51.2      | 60.3        |
> > > | (8, 8), 0       | 29              | 68.1      | 56.2       | 49.7      | 58.0        |
> > > | (2, 4), 4       | 45              | 72.4      | 58.3       | 51.3      | 60.7        |
> > > | (2, 8), 4       | 40              | **72.7**  | 58.2   | **52.9**  | **61.3**    |
> > > | (2, 4), 8       | 41              | 70.1      | 57.6       | 50.8      | 59.5        |
> > > | (2, 8), 8       | 35              | 69.7      | 56.4       | 51.4      | 59.2        |
> > > | (2, 4), 16      | 40              | 68.6      | 57.4       | 51.4      | 59.1        |
> > > | (2, 8), 16      | 31              | 70.3      | 56.3       | 50.7      | 59.1        |
> > >
> > > - **$s_h$**: High-resolution pooling stride for the first frame of each group.
> > > - **$s_l$**: Low-resolution pooling stride for the remaining frames within each group.
> > > - **$K$**: Grouping stride, i.e., the number of frames in each group.
> > >
> > > ### Key Observations:
> > > 1. **Performance vs. Memory Tradeoff**:
> > >    - A smaller pooling stride (e.g., \( (2, 2), 0 \)) retains more information but results in significantly higher memory consumption (73 GB).
> > >    - A larger pooling stride (e.g., \( (8, 8), 0 \)) reduces memory usage to 29 GB but leads to noticeable performance degradation.
> > >
> > > 2. **Effectiveness of Progressive Pooling**:
> > >    - The progressive pooling configuration \( (2, 8), 4 \) achieves the best overall performance (61.3) while reducing memory consumption by **~45%** compared to the baseline \( (2, 2), 0 \).
> > >    - This demonstrates that our method effectively balances information retention and memory efficiency.
> > >
> > > ### Conclusion:
> > > The progressive pooling strategy is carefully designed to address the tradeoff between memory usage and information retention. By preserving fine-grained details in frames and compressing redundant information in adjacent frames, it minimizes information loss while significantly reducing memory consumption. The ablation study further confirms its effectiveness in improving model performance under constrained memory budgets.
> > >
> > > We hope this explanation and the supporting results address your concerns. If you have any further questions or suggestions, we would be happy to discuss them.

---

> > > > ### Author Response · Authors · 2024-12-01
> > > >
> > > > **In response to “The paper may not be entirely fair in comparison with methods that use fewer frames, as a higher number of frames inherently contains more information”.**
> > > >
> > > > Thank you for raising this important point regarding the potential unfairness in comparisons due to differences in the number of input frames. We would like to clarify that our experiments were carefully designed to ensure fairness and to demonstrate the unique contributions of our method.
> > > >
> > > > ### Motivation and Key Contribution:
> > > > The primary motivation of our work is to explore **how to extend LMMs trained on short videos to long video understanding tasks** without requiring additional training or fine-tuning. Our proposed **visual context window extension** achieves this in a training-free manner, enabling models like **LLaVA-OneVision**, which are pre-trained on short videos (i.e., 32 frames), to process long videos (i.e., 512 frames).
> > > >
> > > > The results on multiple long video benchmarks show that our method achieves **state-of-the-art performance**, even surpassing models like **LongVILA**, which are specifically trained on long video datasets.
> > > >
> > > > ### Addressing the Concern:
> > > > We acknowledge that inputting more frames inherently provides access to more information. However, simply increasing the number of input frames for models trained on short videos does not necessarily lead to better performance. In fact, as shown in **Figure 1** of our paper, directly increasing the number of input frames for LMMs trained on short videos often results in **performance degradation** due to challenges in handling the additional contextual information effectively.
> > > >
> > > > To further illustrate this phenomenon, we provide quantitative results for two models, **MiniCPMV2.6 8B**[1] and **LLaVA-OneVision 7B**, in the table below:
> > > >
> > > > | **Model**               | **Setting**                    | **Short** | **Medium** | **Long** | **Overall** |
> > > > |--------------------------|---------------------------------|-----------|------------|-----------|-------------|
> > > > | **MiniCPMV2.6 8B**       | 64 frames                     | 71.3      | 59.4       | 51.8      | 60.9        |
> > > > |                          | 256 frames                    | 69.0      | 55.8       | 51.3      | 58.7        |
> > > > |                          | 256 frames (Ours, Training-Free) | **71.4**  | **59.8**   | **53.8**  | **61.9**    |
> > > > | **LLaVA-OneVision 7B**   | 32 frames                     | 69.3      | 55.1       | 49.7      | 58.2        |
> > > > |                          | 256 frames                    | 64.9      | 53.3       | 50.4      | 56.2        |
> > > > |                          | 256 frames (Ours, Training-Free) | **71.6**  | **59.1**   | **52.2**  | **61.0**    |
> > > >
> > > > ### Key Observations:
> > > > 1. **Performance Degradation with More Frames**:
> > > >    - For both models, directly increasing the number of input frames (e.g., from 64/32 frames to 256 frames) without our method leads to **performance degradation** (e.g., from 60.9 to 58.7 for MiniCPMV2.6 8B and from 58.2 to 56.2 for LLaVA-OneVision 7B).
> > > >
> > > > 2. **Effectiveness of Our Method**:
> > > >    - By applying our **training-free visual context window extension**, the same models achieve **significant performance improvements** when processing 256 frames (e.g., from 58.7 to 61.9 for MiniCPMv2.6 8B and from 56.2 to 61.0 for LLaVA-OneVision 7B).
> > > >
> > > > ### Ensuring Fair Comparisons:
> > > > In our experiments, all baseline methods strictly follow the default settings provided in their respective papers, including the number of input frames. While some baseline methods may use fewer frames, this reflects their original design and training setup, rather than an unfair comparison.
> > > >
> > > > ### Conclusion:
> > > > We hope this explanation clarifies that the comparisons in our paper are fair and that our method's improvements are not merely due to the use of more frames. Instead, they stem from the **training-free adaptation** enabled by our visual context window extension.
> > > >
> > > > If you have further questions or suggestions, we would be happy to address them. Thank you for your thoughtful feedback and for giving us the opportunity to clarify our contributions.
> > > >
> > > > [1] Yao Y, Yu T, Zhang A, et al. MiniCPM-V: A GPT-4V level MLLM on your phone. arXiv preprint arXiv:2408.01800, 2024.

---

> ### Comment · Reviewer_yZzs · 2024-12-01
> **Official Comment by Reviewer**
>
> Thank you for your responses!

---

> > ### Author Response · Authors · 2024-12-01
> >
> > Dear Reviewer #yZzs,
> >
> > Thank you for your kind response!
> >
> > Considering that the rebuttal deadline has been extended to December 2nd, please feel free to reach out if you have any further questions or concerns. We would be more than happy to address them.
> >
> > Best regards,
> >
> > Authors of Paper #2062

---

### Official Review · Reviewer_LsST · 2024-11-03

**Soundness:** 2
**Presentation:** 2
**Contribution:** 2
**Rating:** 5
**Confidence:** 4

**Summary:**

This paper investigates the application of large language models (LLMs) for long video understanding. It first identifies that discrepancies between visual and language modalities lead to different context windows for visual and language tokens. Inspired by the YarN approach, the paper proposes an improvement to the visual context window for video understanding models using RoPE encoding. Additionally, to address the issue of high token count and the associated training and inference overhead for long videos, a progressive pooling strategy is introduced, reducing memory consumption by 45%. The proposed model achieves state-of-the-art results on benchmarks such as MLVU.

**Strengths:**

1. Effective adaptation and optimization of YarN for long video understanding. The paper highlights the need for different context windows for vision and language, enhancing long video comprehension.

2. The progressive pooling strategy with dual compression ratios reduces training and inference costs while retaining dynamic information.

**Weaknesses:**

1. More ablation study and analysis should be done regarding progressive pooling strategy, raising doubts about the role of high-compression frames (see questions 3 and 4).

2. If the answer to Question 1 is "the same context window is used for both language and video parts in video question answering tasks", this part seems more like parameter tuning specific to a particular model and training data setting. It has some contributions but may lack novelty.

**Questions:**

1. Regarding the visual context window extension, some clarification is needed to prevent misunderstanding. In the context of video question-answering, which of the following interpretations is correct?
- Interpretation 1: The visual context window and language context window coexist. In other words, during video question-answering, RoPE encoding for the visual part uses the visual context window for interpolation, while the textual part uses the language context window.
- Interpretation 2: The same visual context window is used for interpolation for both the visual and textual parts during video question-answering.

If Interpretation 1 is correct, how is attention computed between the visual and textual parts given the difference in relative position encoding?

2. According to the citation of allava dataset, it seems to be an image dataset. How can it be used to fine-tune the video understanding model in the paper?

3. In the ablation study of progressive pooling strategy, why does (2,4),4 perform better in some cases compared to (2,2),0?

4. Why does (2,8),4 outperform (2,4),4? Moreover, given that 8 is a very high compression ratio, if the ViT patch size is 14×14, this would compress a 224×224 image frame down to a length of $224/14/8=2$ per side, equating to 4 tokens. Since inter-frame pooling is independent, there is a possibility that dynamic information is not captured.
Could it be that the highly compressed frames are actually not contributing? A simple experiment would be, comparing with (2,2), 0 with 64 sampled frames.

5. Will the code and model checkpoints be open-sourced?

If all the questions above are solved, I would raise my score.

---

> ### Author Response · Authors · 2024-11-21
>
> Thank you so much for the thoughtful questions and suggestions. We hope that our response below will address your concerns.
>
> **Q1**: … In the context of video question-answering, which of the following interpretations is correct?
>
> **R1**: Thank you for your suggestion. In video question-answering tasks, both the visual and textual components share the same visual context window for interpolation. We conducted experiments on both approaches and observed that when the visual component uses the visual context window for interpolation while the textual component uses the language context window, the model fails to generate complete words.
> We believe this issue arises because the relative positional relationships between visual tokens and language tokens are disrupted. Considering that textual sequences in video question-answering tasks are typically short, and to ensure our method remains simple and easy to integrate into existing training and deployment frameworks, we opted to use the same visual context window for both visual and textual components. In fact, implementing visual context window extension only requires modifying a few lines of code in existing open-source models. This allows for extending LMMs trained on short videos to handle long videos effectively.
>
> **Q2**: According to the citation of allava dataset, it seems to be an image dataset. How can it be used to fine-tune the video understanding model in the paper?
>
> **R2**: Thank you for your suggestion. In our experiments, we used the LLava-OneVision backbone, which supports both image and video understanding tasks by treating videos as sequences of sampled frames. The reason we utilized the ALLaVA image dataset is to align with the methodology of LongVA [1], which achieves long video understanding by training solely on images.
> Fine-tuning on an image dataset also reduces computational resource requirements. For example, our approach only requires an RTX 3090 24G GPU to fine-tune the model, which can be particularly advantageous in resource-constrained scenarios. Moreover, using an image dataset aligns with the core motivation of this paper: achieving strong performance on long video understanding tasks without fine-tuning the model on long video data.
>
> [1] Peiyuan Zhang, Kaichen Zhang, Bo Li,et.al. Long context transfer from language to vision.
> arXiv preprint arXiv:2406.16852, 2024.
>
> **Q3**: In the ablation study of progressive pooling strategy, why does (2,4),4 perform better in some cases compared to (2,2),0?
>
> **R3**: Thank you for your suggestion. In the ablation study of the progressive pooling strategy, we observed that (2,4),4 outperforms (2,2),0 on short videos. We believe this phenomenon is caused by **attention distraction**. When densely sampling short videos (e.g., 256 frames), a large number of redundant tokens are introduced.
> Previous studies [1][2] have shown that excessive context irrelevant to the query can lead to attention distraction, resulting in a significant drop in performance. This issue is even more pronounced in long-video understanding tasks, where video content tends to be highly redundant, and the relevant segments may only appear for a few seconds.
> During inference, when the input is fixed at 256 frames, the number of visual tokens under the progressive pooling strategy (2,4),4 is 21,952, whereas for the (2,2),0 strategy, the number of visual tokens is 50,176. The significantly larger number of tokens in the latter increases redundancy and exacerbates attention distraction, which explains the performance difference.
>
> [1] Shi F, Chen X, Misra K, et al. Large language models can be easily distracted by irrelevant context. ICML, 2023: 31210-31227.
>
> [2] Li Z, Zhang Y, Pan T, et al. FocusLLM: Scaling LLM's Context by Parallel Decoding. arXiv preprint arXiv:2408.11745, 2024.

---

> > ### Author Response · Authors · 2024-11-21
> >
> > **Q4**: Why does (2,8),4 outperform (2,4),4? … Could it be that the highly compressed frames are actually not contributing? A simple experiment would be, comparing with (2,2), 0 with 64 sampled frames.
> >
> > **R4**: Thank you for your suggestion. Regarding why (2,8),4 outperforms (2,4),4, we believe this is due to the smaller pooling stride in (2,4),4 introducing more redundant visual information. This results in longer input visual sequences, which can lead to **attention distraction** and a subsequent drop in model performance.
> > When videos are sampled continuously, adjacent frames often share similar backgrounds, leading to repetitive visual tokens. This redundancy increases the length of the input sequence and amplifies the problem of redundant information.
> >
> > In our experiments, we used LLava-OneVision as the backbone, which employs SigLIP as the visual encoder. For input video frames, the number of visual tokens under different pooling strides is as follows:
> > - **Pooling stride = 0:** 27×27 tokens
> > - **Pooling stride = 2:** 14×14 tokens
> > - **Pooling stride = 4:** 7×7 tokens
> > - **Pooling stride = 8:** 4×4 tokens
> >
> > Following your suggestion, we compared (2,2),0 with 64 sampled frames on the VideoMME dataset. The results are shown in the table below:
> >
> > | Method       | Short | Medium | Long | Overall |
> > |--------------|------------|------------|------------|------------------|
> > | 64 frames     | 72.3       | 58.0       | 50.6       | 60.3             |
> > | (2,2),0     | 71.6       | 59.1       | 52.2       | 61.0             |
> >
> > The results indicate that (2,2),0 outperforms the 64-frame baseline, suggesting that even highly compressed frames contribute to long video question-answering tasks.
> > Additionally, previous works, such as **LLaMA-VID** [1], have explored frame compression in video understanding. Llama-vid demonstrated that even using only 2 tokens to represent a video frame can effectively contribute to video understanding.
> >
> > [1] Li Y, Wang C, Jia J. Llama-vid: An image is worth 2 tokens in large language models. ECCV, 2025: 323-340.
> >
> > **Q5**: Will the code and model checkpoints be open-sourced?
> >
> > R5: We are committed to open-sourcing the code and model checkpoints in the near future, contributing to the open-source community for long-video understanding.

---

> ### Comment · Reviewer_LsST · 2024-11-22
>
> Thank you for your response. However, I still have some concerns regarding the limited novelty of the paper, so I decide to keep my original score.

---

> > ### Author Response · Authors · 2024-11-25
> >
> > Dear Reviewer #LsST,
> >
> > Thank you very much for your tremendous effort and valuable feedback on our paper. While we regret that we were unable to reach a consensus, we sincerely appreciate your comments, as they are crucial for improving the quality of our work.
> >
> > We would like to make a final statement regarding the novelty of our work:
> >
> > 1. **Key Discovery**:
> >
> >     In this paper, we are the first to identify the issue that **differences between the vision and language modalities lead to performance degradation when directly extrapolating visual token positional embeddings to the language context window**. This is a novel observation that has not been explicitly addressed in prior work.
> >
> > 2. **Proposed Solution**:
> >
> >     Based on this finding, we propose the visual context window extension method, which offers a new perspective for addressing long video understanding tasks. Our method enables the extension of LMMs trained on short videos (i.e., **32** frames) to handle long videos (i.e., **1024** frames) **without requiring additional training**. This approach achieves state-of-the-art (SoTA) performance and could provide significant benefits in resource-constrained scenarios.
> >
> >     Additionally, our method does not involve modifying the underlying model architecture, which ensures that it can continuously benefit from future advancements in LMMs. This adaptability highlights the potential impact of our work in both research and practical applications.
> >
> > 3. **Memory Efficiency**:
> >
> >     We also propose a progressive pooling strategy, which, in the 256-frame setting, reduces memory usage by approximately **45%** compared to the baseline without introducing any performance loss. This improvement demonstrates the practicality of our approach for real-world applications where computational resources are limited.
> >
> > 4. **Simplicity and Usability**:
> >
> >     While our method is simple, its simplicity ensures **ease of implementation and deployment**. In many cases, simple yet effective solutions are more likely to be adopted in real-world scenarios, which further underscores the practical value of our work.
> >
> > Once again, we sincerely thank you for taking the time to review our paper and for providing constructive feedback. While we may not have reached an agreement on the novelty of our work, your comments have been invaluable in helping us refine our research and presentation.
> >
> > Thank you for your consideration.

---

> > > ### Author Response · Authors · 2024-11-26
> > >
> > > **Question**: If the answer to Question 1 is "the same context window is used for both language and video parts in video question answering tasks", this part seems more like parameter tuning specific to a particular model and training data setting. It has some contributions but may lack novelty.
> > >
> > > **Response**: We sincerely apologize for overlooking this question in our initial response, which may have led to the misunderstanding regarding the perceived novelty of our work.
> > >
> > > To clarify, the proposed visual context window extension is **not a parameter tuning approach specific to a particular model or training data setting**. Instead, it is a general method that can be applied to any LMMs using RoPE (Rotary Position Embedding), which is one of the most commonly used positional encoding techniques in both LMMs and LLMs. Importantly, our method is independent of training data and serves as a **training-free solution** for extending LMMs to long video understanding tasks.
> > >
> > > The fine-tuning we performed on the ALLaVA dataset was solely for validation purposes, demonstrating that fine-tuning can further enhance the performance of our method. However, as shown in the experimental section, except for the results in **Table 4**, all other experiments were conducted in a **training-free setting**, emphasizing the generalizability and practicality of our approach.
> > >
> > > Furthermore, to support the research community and facilitate progress in long video understanding tasks, we have **provided the code for our method in the supplementary materials**. We hope this will contribute to the development of open-source tools and inspire further research in this area.
> > >
> > > If you have any additional concerns or suggestions, please feel free to reach out. We greatly appreciate your time and effort in reviewing our work.

---

> > > > ### Author Response · Authors · 2024-12-01
> > > >
> > > > **In response to “limited novelty”**
> > > >
> > > > We would like to address the concern that it "**seems more like parameter tuning specific to a particular model and training data setting. It has some contributions but may lack novelty**" by providing additional evidence to demonstrate the generalizability and effectiveness of our approach.
> > > >
> > > > Our method is **not a parameter tuning approach specific to any particular model or dataset**, but rather a **general and training-free solution** that can be applied to extend the context length of existing LMMs for long video understanding tasks.
> > > >
> > > > To further validate the universality of our method, we conducted experiments on an additional model, **MiniCPMV2.6 8B** [1], in addition to **LLaVA-OneVision 7B**. The results are summarized below:
> > > >
> > > > | **Model**               | **Setting**                 | **Short** | **Medium** | **Long** | **Overall** |
> > > > |--------------------------|-----------------------------|-----------|------------|-----------|-------------|
> > > > | **MiniCPMV2.6 8B**       | 64 frames                  | 71.3      | 59.4       | 51.8      | 60.9        |
> > > > |                          | 256 frames                 | 69.0      | 55.8       | 51.3      | 58.7        |
> > > > |                          | 256 frames (Ours, Training-Free) | **71.4**  | **59.8**   | **53.8**  | **61.9**    |
> > > > | **LLaVA-OneVision 7B**   | 32 frames                  | 69.3      | 55.1       | 49.7      | 58.2        |
> > > > |                          | 256 frames                 | 64.9      | 53.3       | 50.4      | 56.2        |
> > > > |                          | 256 frames (Ours, Training-Free) | **71.6**  | **59.1**   | **52.2**  | **61.0**    |
> > > >
> > > > As shown in the table, our method consistently improves performance across different backbone models (MiniCPMV2.6 8B and LLaVA-OneVision 7B) in a training-free setting. These results demonstrate that the visual context window extension is a **model-agnostic and generalizable technique** that is not limited to any specific model or dataset.
> > > >
> > > > We hope this additional evidence addresses your concerns regarding the novelty of our approach. If you have any further questions or suggestions, please do not hesitate to let us know. We deeply appreciate your thoughtful feedback and the opportunity to clarify the contributions of our work.
> > > >
> > > > [1] Yao Y, Yu T, Zhang A, et al. Minicpm-v: A gpt-4v level mllm on your phone. arXiv preprint arXiv:2408.01800, 2024.

---

### Comment · Area_Chair_5HeD · 2024-11-28

Dear reviewers,

This is a friendly reminder that the discussion period has been extended until December 2nd. If you haven’t yet, we kindly encourage you to review the authors' rebuttal and messages at your earliest convenience and confirm whether your comments have been adequately addressed.

We greatly appreciate your service to this process.

Best, AC

---

### Meta-Review · Area_Chair_5HeD · 2024-12-19

**Metareview:**

This paper tackles the challenge of long video understanding by redefining the context window of LMMs into visual and language components, enabling the use of a visual context window extension to apply LMMs trained on short videos to long video tasks without fine-tuning.  The final scores are 5, 5, 6, 6.  Strengths include good motivation, finetuning process not requiring long video datasets, and promising results.  Weaknesses include limited novelty, limited experiments, lacking theoretical insights, some issues with the approach, and weak results on short video datasets.  The rebuttal addressed several of these concerns, including evaluation on interleaving of visual and text features, the key benefit of the visual context length extension, theoretical insights, and good results with 256 frames, which led to a reviewer increasing their score.  However, some concerns, including limited novelty were not adequately addressed.  The AC carefully reviewed the paper, rebuttal, and messages between the authors and reviewers.  Unfortunately, the AC feels that the weaknesses outweigh the strengths, and that the paper is not ready for publication.  The AC encourages the authors to revise and improve the paper and resubmit to a future conference.

**Additional Comments On Reviewer Discussion:**

Strengths include good motivation, finetuning process not requiring long video datasets, and promising results.  Weaknesses include limited novelty, limited experiments, lacking theoretical insights, some issues with the approach, and weak results on short video datasets.  The rebuttal addressed several of these concerns, including evaluation on interleaving of visual and text features, the key benefit of the visual context length extension, theoretical insights, and good results with 256 frames, which led to a reviewer increasing their score.  However, some concerns, including limited novelty were not adequately addressed.  The AC carefully reviewed the paper, rebuttal, and messages between the authors and reviewers.  Unfortunately, the AC feels that the weaknesses outweigh the strengths, and that the paper is not ready for publication.  The AC encourages the authors to revise and improve the paper and resubmit to a future conference.

---

### Decision · Program_Chairs · 2025-01-22

Reject